# Unsupervised Learning of Temporal Abstractions with Slot-based Transformers

## Abstract

The discovery of reusable sub-routines simplifies decision-making and planning in complex reinforcement learning problems. Previous approaches propose to learn such temporal abstractions in an unsupervised fashion through observing state-action trajectories gathered from executing a policy. However, a current limitation is that they process each trajectory in an entirely *sequential* manner, which prevents them from revising earlier decisions about sub-routine boundary points in light of new incoming information. In this work we propose SloTTAr, a fully parallel approach that integrates sequence processing Transformers with a Slot Attention module to discover sub-routines in an unsupervised fashion, while leveraging adaptive computation for learning about the number of such sub-routines solely based on their empirical distribution. We demonstrate how SloTTAr is capable of outperforming strong baselines in terms of boundary point discovery, even for sequences containing variable amounts of sub-routines, while being up to 7x faster to train on existing benchmarks.

## 1 Introduction

An intelligent goal-seeking agent situated in the real world should make decisions along various timescales to act and plan efficiently. A natural approach that facilitates this behavior is via a divide-and-conquer strategy, where goals are decomposed along sub-goals, and solutions to such sub-goals (i.e. 'correct' sequences of actions) are stored as reusable 'primitive' *sub-routines* (Schmidhuber, 1991; Dayan & Hinton, 1992; Bakker & Schmidhuber, 2004; Schmidhuber, 1990). Viewing complex goal-directed behavior as (novel) compositions of known sub-routines simplifies both decision-making and planning (Schmidhuber & Wahnsiedler, 1993; Tadepalli & Dietterich, 1997), while also benefiting out-of-distribution generalization (Peng et al., 2019).

The options framework (Sutton et al., 1999) introduced a design strategy for the hierarchical organization of behavior within the context of reinforcement learning. Crucial to its success is the quality of each 'option' (or sub-routine) in terms of the level of *modularity* and *reusability* it offers. Consider for example, the task of planning a travel itinerary to go from London to New York. Useful sub-routines in this context may include purchasing flight tickets, navigating to the airport, or boarding the flight. This factorization into sub-routines is desirable as these sub-routines capture mostly *self-contained* activities. Thus they are far more likely to apply to other travel plans between two different cities, which may again involve taking flights, etc.

Prior approaches propose to address this issue by learning about useful sub-routines directly from data (Andreas et al., 2017; Shiarlis et al., 2018; Kipf et al., 2019; Lu et al., 2021). Of particular interest is the setting where the model is only given access to state-action trajectories from an (expert) policy and learns in an unsupervised manner (while the ground-truth number of sub-routines are assumed to be known). Two relevant works can be distinguished: CompILE (Kipf et al., 2019) and Ordered Memory Policy Network (OMPN) (Lu et al., 2021). In CompILE, expert trajectories are modeled using a latent variable model based on a recurrent neural network (RNN), which infers a pre-determined number of boundary points by iteratively processing the entire trajectory multiple times to recover segments associated with reusable sub-routines. Alternatively, OMPN equips an RNN with a multi-layer hierarchical memory, where information processing at different levels in the memory can be interpreted as belonging to separate segments. While OMPN additionally includes a strong preference for capturing hierarchical relationships between different sub-routines, both

methods are limited insofar that they process the trajectory in an entirely *sequential* manner. This prevents them from revising earlier decisions about boundary points in light of new information that becomes only accessible at a future stage. Moreover, iteratively processing the sequence multiple times (as in CompILE) or interfacing with a deep hierarchical memory (as in OMPN) incurs significant computational costs.

In this paper we propose a novel architecture to learning sub-routines that addresses these shortcomings, which we call *Slot-based Transformer for Temporal Abstraction (SloTTAr)*. Central to our approach is the similarity between the *spatial grouping* of pixels into visual objects and the *temporal grouping* of state-action pairs into self-contained sub-routines. This motivates us to combine a parallel Transformer encoder with a Slot Attention module (Locatello et al., 2020) (developed for learning object representations c.f. Greff et al. (2020)), to group learned features at different temporal positions and recover a modular factorization into 'slots' (sub-routines) of the inputs. A parallel Transformer decoder reconstructs the action sequence from each slot and outputs an unnormalized distribution over endpoints. From these, the segment belonging to each sub-routine and a standard reconstruction objective for unsupervised training can be derived.

In more realistic scenarios, coping with a variable number of sub-routines per trajectory adds an additional layer of complexity in order to achieve an ideal "fully unsupervised" sequence decomposition (i.e. without the knowledge about the number of sub-routines). In CompILE and OMPN, this issue is not addressed as the number of sub-routines in each trajectory is assumed to be known in advance, either at training and/or at test time, which limits their applicability. In contrast, in SloTTAr we cast the problem of estimating the number of sub-routines as a form of *adaptive computation* (where each available slot corresponds to an available computational step), requiring it to learn to use only some adaptive subset of all the slots available based on an estimated prior. This motivates us to use the adaptive computation loss introduced in PonderNet (Banino et al., 2021) as our learning objective. It allows for only an adaptive subset of the available slots in SloTTAr to be actively involved in the decomposition of any trajectory. While SloTTAr is still not "fully unsupervised", it only requires a weak supervision for the problem of predicting the number of sub-routines in the form of their empirical distribution during training, and unlike CompILE and OMPN, it does not require any such information at test time.

We demonstrate the efficacy of our approach on Craft (Lu et al., 2021) and Minigrid (Chevalier-Boisvert et al., 2018) where we typically observe significant improvements over CompILE and OMPN in terms of recovering 'ground-truth' sub-routines (Sections 4.4.1 and 4.4.2). In this way, we show how general principles of similarity-based grouping used to segment visual inputs into objects (Spelke, 1990; Köhler, 1929; Koffka, 1935; Greff et al., 2020) are also relevant for grouping other input modalities (Hommel, 1998). Our qualitative and quantitative analysis reveals how SloTTAr leverages global access to the full trajectory to better capture individual sub-routines (Section 4.4.4) and leads to a substantial speed-up on existing benchmarks (Section 4.4.5). Finally, we demonstrate how our model is capable of modeling trajectories, each of which constituted of a variable number of sub-routines, without requiring access to ground-truth information about this quantity at test-time (Section 4.4.3).

## 2 Method

Given an input sequence of actions $\boldsymbol{a} = [a_1, a_2, ..., a_L] : a_l \in \{1, .., A\}$ where $A$ is the size of the action space and observations $\boldsymbol{o} = [\boldsymbol{o}_1, \boldsymbol{o}_2, ..., \boldsymbol{o}_L] : \boldsymbol{o}_l \in \mathbb{R}^{D_{\text{obs}}}$ of length $L$. Our goal is to infer the unique constituent latent *sub-routine* to which a pair of $(a_l, \boldsymbol{o}_l)$ belongs, and learn its associated representation ($\texttt{slot\_k} \in \mathbb{R}^{D_{\text{slots}}}$). The input sequence of actions $\boldsymbol{a}$ is assumed to be a composition of at most $K$ sub-routines. Similar to CompILE (Kipf et al., 2019), we consider an unsupervised reconstruction objective for training, yet here we strive for a fully parallel approach that employs more general-purpose modules for grouping inputs based on their internal predictive structure.

Our model, which we call *Slot-based Transformer for Temporal Abstraction (SloTTAr)*, consists of three modules, as shown in Figure 1. First, a Transformer encoder (Vaswani et al., 2017) learns suitable spatio-temporal features from the input sequences $\boldsymbol{a}$ and $\boldsymbol{o}$. This encoder benefits from *global* access to the whole input sequence to learn suitable context features at each location, unlike prior purely sequential RNN-based approaches (Kipf et al., 2019; Lu et al., 2021). Subsequently, we adapt the Slot Attention module (Locatello et al., 2020) to iteratively group the features at each temporal location in parallel based on their internal predictive structure and obtain $K$ slot representations $\texttt{slot\_k}$. The slot representations are decoded in

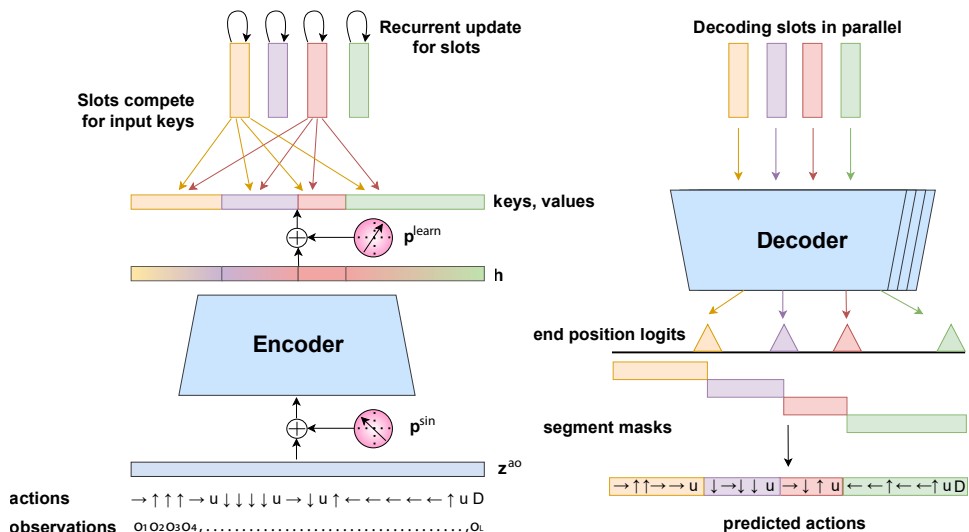

Figure 1: SloTTAr consists of 3 modules, used for: learning spatio-temporal features of action-observation sequences (*Encoder*), learning a similarity-based grouping of actions to their respective sub-routines through computing slot-based representations (*Slot Attention*; Locatello et al. (2020)), and for decoding slots to end positions and action segments of the sub-routine they capture (*Decoder*).

parallel by a decoder that outputs both predicted action logits $\hat{\boldsymbol{a}}^{(k)}$ and an unnormalized distribution over the end position of the sub-routine `end_logits_k`, modeled by the respective slot $k$.

Sub-routine segment masks are generated from the unnormalized distributions and combined with the predicted action logits (across slots) to obtain aggregated action sequence logits $\hat{\boldsymbol{a}}$. For training, we use the PonderNet loss (Banino et al., 2021) as our learning objective which comprises two terms. The first being a weighted sum of reconstruction errors of the action sequence, where each term corresponds to the error obtained when using $k$ slots, and the weight is the probability that the trajectory contains $k \in \{1, \dots, K\}$ sub-routines. The second is a regularization cost given by a KL-divergence term between the posterior and prior distributions of the number of sub-routines in a trajectory.

**Encoder.** We process action tokens $\boldsymbol{a}$ and observations $\boldsymbol{o}$ by `Embedding` and `Linear` layers respectively to acquire separately learned distributed representations for each. Next, we learn a joint representation of the action and observation representations at each position using a one-layer `MLP` with `ReLU` activation, after which a sinusoidal positional encoding $p_l^{\text{sin}} \in \mathbb{R}^{D_{\text{enc}}}$ is added (Vaswani et al., 2017). Finally, we apply a number of standard Transformer encoder layers to learn suitable spatio-temporal features based on the content of the *entire* sequence, and add a learned positional encoding $p^{\text{learn}} \in \mathbb{R}^{D_{\text{enc}}}$ for follow-up processing:

$$\boldsymbol{z}_l^a, \ \boldsymbol{z}_l^o = \texttt{Embedding}(a_l), \ \texttt{Linear}(\boldsymbol{o}_l) \qquad \boldsymbol{z}_l^{ao} = \texttt{MLP}(\texttt{concat}(\boldsymbol{z}_l^a, \boldsymbol{z}_l^o)) \qquad \forall \, l \in \{1, \dots, L\}$$

$$\boldsymbol{h} = \texttt{TransformerEnc}(\boldsymbol{z}^{ao} + \boldsymbol{p}^{\text{sin}}) + \boldsymbol{p}^{\text{learn}}$$

**Slot Attention.** We adapt Slot Attention (Locatello et al., 2020) to group these spatio-temporal features according to their constituent sub-routines and learn associated representations given by the slots. Slot Attention was previously only applied to the visual setting where pixels are grouped according to constituent visual objects to model images or videos. Here we hypothesize that sub-routines take on a similar role as modular and reusable primitives when modeling action sequences, suggesting that they similarly can be inferred using a grouping mechanism that focuses on their internal predictive structure (modularity) (Greff et al., 2020).

The iterative grouping mechanism in Slot Attention (reproduced in Algorithm 1) is implemented via key-value attention and a stateful slot update rule using a recurrent neural network (GRU; Cho et al. (2014), see also

---

**Algorithm 1** Modified Slot Attention update (Locatello et al., 2020).

Our modification uses separate shift and scale parameters per-slot (highlighted in blue) to initialize each slot representation `slot_k`.

---

**Inputs:** `inputs` $\in \mathbb{R}^{L \times D_{\text{in}}}$, `slots` with `slot_k` $= \boldsymbol{\mu}_k + \boldsymbol{\sigma}_k * \boldsymbol{z}$, $\boldsymbol{z} \sim \mathcal{N}(\mathbf{0}, diag(\sigma)) \in \mathbb{R}^{D_{\text{slots}}} \; \forall k \in \{1, \ldots, K\}$

**Params:** shift and scale parameters : $\boldsymbol{\mu}_k$, $\boldsymbol{\sigma}_k$ per-slot; Linear projections for attention: $key, query, value$; GRU; MLP; LayerNorm x2

1: **for** $t = 0, 1, ..., T$ **do**
2:     `slots_prev = slots`
3:     `slots = LayerNorm(slots)`
4:     `attn = Softmax(`$\frac{1}{\sqrt{D_{\text{slots}}}}$ $key$`(inputs)`$\cdot$ $query$`(slots)`$^{\text{T}}$`, axis='slots')`         ▷ norm. over $K$
5:     `updates = WeightedMean(weights=attn+`$\epsilon$`, values=`$value$`(inputs))`  ▷ norm `attn` values over $L$
6:     `slots = GRU(state=slots_prev, inputs=updates)`         ▷ GRU update (per-slot)
7:     `slots += MLP(LayerNorm(slots))`         ▷ residual update (per-slot)
8: **end for**
9: **return** `slots`

---

the earlier work by Gers et al. (2000)). The input features are projected to keys and values using linear layers $key(\cdot)$, $value(\cdot)$, queries are computed from slots using a linear layer $query(\cdot)$, and dot-product attention between keys and queries is used to distribute value vectors among the slots. Initial representations for all slots are sampled from a Gaussian $\mathcal{N}(\mathbf{0}, diag(\sigma))$ where $\sigma$ is a hyperparameter. Here, unlike in the original Slot Attention formulation, we additionally use separate shift and scaling parameters ($\boldsymbol{\mu}_k$ and $\boldsymbol{\sigma}_k$) for each slot[1]. This allows a slot to differentiate itself from others and thus to focus on a specific part of the input sequence in the first iteration (e.g. by targeting the positional embedding). Importantly, slots *compete* to represent parts of the input sequence via a softmax function (`Softmax`) over slots, thereby encouraging a decomposition of the input into modular parts that can be processed separately and represented efficiently.

**Decoder.** To decode the slot representations and obtain a correspondence of actions in **a** to their constituent sub-routine segments, we adapt the spatial broadcast decoder architecture (Watters et al., 2019) to the sequential setting using Transformers. Each slot representation is decoded in parallel to obtain the predicted action logits $\hat{\mathbf{a}}^{(k)}$ as well as an unnormalized distribution over the end position of the sub-routine in the sequence (`end_logits_k`). The latter is used to generate the segment masks corresponding to each sub-routine as described in Algorithm 2. Algorithm 2 imposes that each slot takes responsibility for modeling a contiguous sub-sequence of the input. This is because mask generated by `slot_k+1` is active (value of 1) for all the time steps from the endpoint of that of `slot_k` to its own predicted end point. This is in contrast to the standard mixture formulation adopted by prior work on discovering visual objects (Greff et al., 2017; 2019; Locatello et al., 2020). This has a similar effect to the sequential decoding in CompILE (Kipf et al., 2019) and provides a useful inductive bias for treating sub-routines as contiguous chunks in time that can be ordered along the temporal axis, while keeping all other computations parallel.

**Objective Function.** We adopt the PonderNet (Banino et al., 2021) adaptive computation loss formulation to model the variable number of sub-routines contained in different trajectories. We define a Bernoulli random variable $\Lambda_k$ which represents a Markov process for halting with two states "continue" ($\Lambda_k = 0$) and "halt" ($\Lambda_k = 1$). The decision process always starts from the "continue" state and the transition probability is given by:

$$P(\Lambda_k = 1 | \Lambda_{k-1} = 0) = \lambda_k, \qquad \forall 1 \leq k \leq K. \tag{1}$$

where $\lambda_k$ is the probability of halting at the $k^{th}$ `slot`, which is computed by applying a sigmoid activation to the last dimension of its respective slot representation `slot_k`. Then, the total probability that halting

---

[1]A similar modification was explored as an ablation in (Locatello et al., 2020).

---

**Algorithm 2** Mask Generation

`CumSum()` is a function that computes the cumulative sum of the input array it receives along the specified dimension (here the time-axis 'L'). Further, `mask_upto_k` is a variable that is a union over masks generated by all slots up to the $k^{th}$ while `mask_upto_km1` is an analogous quantity except that is up to the $k - 1^{th}$ slot. For a visualization of the mask generation process, we refer the reader to Figure 6.

---

**Inputs:** `end_logits` $\in \mathbb{R}^{K \times L}$, `masks = []`, `end_cdf_k = 1`, `mask_upto_km1 = 0`

1: **for** $k = 1, \ldots, K$ **do**
2:     `end_dist_k = Softmax(end_logits_k, axis='L')`          ▷ norm. over $L$
3:     `end_cdf_k = CumSum(end_dist_k, axis='L')`          ▷ end position CDF
4:     `mask_upto_k = 1 - end_cdf_k`          ▷ union of all masks upto k
5:     `mask_k = mask_upto_k * (1 - mask_upto_km1)`          ▷ compute $k^{th}$ mask
6:     `mask_upto_km1 = mask_upto_k`          ▷ update
7:     `masks = Append(masks, mask_k)`          ▷ append $k^{th}$ mask
8: **end for**
9: **return** `masks`

---

occurs in step $k \in \{1, \ldots, K\}$ (and the induced distribution $p_{\text{halt}}$) is given by:

$$p_k = \lambda_k \prod_{j=1}^{k-1} \left(1 - \lambda_j\right), \qquad p_{\text{halt}}(k) = \frac{p_k}{\sum\limits_{k'=1}^{K} p_{k'}}, \tag{2}$$

where the normalization on the right ensures that $p_{\text{halt}}$ is a valid probability distribution. At test time, we determine the number of 'active' slots by sampling for the $k^{th}$ `slot` from a Bernoulli distribution $\mathcal{B}(\lambda_k)$ and proceeding until we receive a halting signal. The entire system is trained end-to-end to minimize the following objective function:

$$\mathcal{L} = \sum_{k=1}^{K} p_{\text{halt}}(k) \mathcal{L}_{\text{CE}}\left(\hat{\mathbf{a}}^{(\leq k)}; \mathbf{a}\right) + \beta D_{KL}\left[p_{\text{halt}} || p_e\right], \tag{3}$$

where $\mathcal{L}_{\text{CE}}$ refers to the standard cross-entropy loss between the reconstruction $\hat{\mathbf{a}}^{(\leq k)}$ obtained by using up to $k$ slots (i.e., $\hat{\mathbf{a}}^{(\leq k)} = \sum_{k'=1}^{k} \text{mask\_k}' * \hat{\mathbf{a}}^{(k')}$) and the ground-truth action sequence $\mathbf{a}$, and $\beta$ is a hyperparameter that weights the influence of the second term. The second term is a KL-divergence between the probability distribution induced by $p_{\text{halt}}$ (Equation (2)) and a prior distribution $p_e$ governing the expected number of sub-routines. The empirical prior distribution $p_e$ is derived from the histogram of the number of sub-routines in each trajectory across the training dataset. Note that this requires only a weaker form of ground-truth information about the *distribution* of the number of sub-routines in the training data (compared to CompILE), while no such information is needed at test-time (compared to OMPN). Please refer to Appendix A.3 for additional details about the model architecture and the estimated empirical prior distribution.

## 3    Related Work

**Imitation Learning of Temporal Abstractions.** The Craft environment was originally introduced to evaluate the method of Andreas et al. (2017), however they rely on annotations of sub-routine sequences ('policy sketches') as additional supervision to the model. In a similar way, TACO (Shiarlis et al., 2018) treats this setting as a sequence alignment and classification problem using an LSTM (Hochreiter & Schmidhuber, 1997) trained with a CTC loss (Graves et al., 2006). In a recent work of Ajay et al. (2021), the primitives are learned directly from offline data for continuous control. However, these methods learn a continuous (low-dimensional) space of primitives whereas our method represents primitives as a discrete set. Our focus on learning useful sub-routines can also be viewed as an instance of learning temporal abstractions more broadly, such as event segments in video. Relevant approaches propose to use recurrent latent variable models for this task (Gregor et al., 2019; Kim et al., 2019), while making stronger assumptions regarding prior knowledge about boundary locations and the existence of hierarchical structure between latent states and across time.

**Option Discovery.** In the context of the options framework (Sutton et al., 1999), several methods have been proposed for option and/or sub-goal discovery (Bacon et al., 2017; Machado et al., 2017; McGovern & Barto, 2001; Stolle & Precup, 2002; Şimşek & Barto, 2004; Şimşek et al., 2005; Şimşek & Barto, 2008). However, these algorithms are not easily extendable to the case with nonlinear function approximation as needed in our case. Alternatively, other methods using nonlinear function approximation propose to maximize coverage (diversity) of learned skills by maximizing the mutual information between options and terminal states achieved by their execution (Gregor et al., 2017; Eysenbach et al., 2019). It remains difficult to precisely evaluate the level of semantically independent modes of behavior that options discovered in this way afford. In contrast, here we explicitly optimize and evaluate the sub-routines with regards to their modularity.

**Transformer-based Sequence Modeling Beyond Language.** Many recent works have applied Transformers beyond the domain of natural language, such as to images (Dosovitskiy et al., 2021; Zhu et al., 2021; Singh et al., 2022), and other high-dimensional data (Jaegle et al., 2021b;a). Transformers were successfully applied to reason about visual primitives, such as object representations (Ding et al., 2021), and have recently gained a lot of interest in reinforcement learning settings (Parisotto et al., 2020; Rae et al., 2020; Fan et al., 2020; Irie et al., 2021; Chen et al., 2021a; Zeng et al., 2022). Especially in the offline setting (Janner et al., 2021; Chen et al., 2021b), which closely relates to the setting explored here, Transformers have shown promise in predicting actions from certain input commands in a supervised manner (Schmidhuber, 2019; Srivastava et al., 2019).

**Adaptive Computation.** The idea of allowing dynamic program halting behavior in recurrent neural networks was initially explored in Schmidhuber (2012). Here, a special 'halting' unit, when activated, resulted in termination of the program and emission of program outputs. Similarly, the Adaptive Computation Time (ACT) algorithm (Graves, 2016) uses a sigmoidal 'halting unit' (and associated weights) for halting program execution, which is regularized to penalize the amount of computation used. The ACT algorithm for halting has later been adapted to the Transformer family of models as well (Dehghani et al., 2019). PonderNet (Banino et al., 2021) builds on ACT but crucially differs in its use of a probabilistic halting strategy. It defines a probability distribution over all possible computation steps available and weights the output prediction loss at each step with their associated halting probability. Further, PonderNet uses a KL-divergence between the posterior distribution over the number of computation steps and a prior distribution as a regularizer.

**Event Cognition.** Influential work from cognitive psychology on object perception in humans (Kahneman et al., 1992) posit that the brain integrates distributed representations of perceptual (visual) stimuli into *object files*. Further studies have suggested that such a feature integration mechanism extends beyond just the visual stimuli and include associated behavioral responses (actions) as well (Hommel, 1998; 2004; 2007). These works suggest that the definition of an *object files* could be extended to include a notion of *event files* which store aggregated information of action-perception stimuli. More specifically, our ability to chunk up a continuous stream of activity into a set of semantically meaningful *event entities* depends on both bottom-up sensory cues of color, sound, movement etc. as well as top-down cues like goals and beliefs (Radvansky & Zacks, 2014). Further, these events tend to have hierarchical organization (Zacks et al., 2001; Hard et al., 2006) and play an important role in decision-making and planning (Radvansky & Zacks, 2014). The Slot Attention module in SloTTAr analogously integrates information by attending to bottom-up features of both perceptual (observations) and behavioral (actions) stimuli to learn the appropriate slot representations for sequence decomposition.

## 4 Experiments

We compare SloTTAr to CompILE (Kipf et al., 2019) and OMPN (Lu et al., 2021) on the partially and fully observable versions of environments in Craft (Lu et al., 2021) and on partially observable environments in Minigrid (Chevalier-Boisvert et al., 2018). Here our focus is on the unsupervised setting without the use of task sketches as an auxiliary supervision signal (Andreas et al., 2017; Shiarlis et al., 2018).

### 4.1 Datasets

We use 3 tasks in the Craft environment namely, `MakeAxe, MakeBed, MakeShears` consistent with prior work (Lu et al., 2021). We also compare models on 4 tasks in Minigrid suite of environments namely

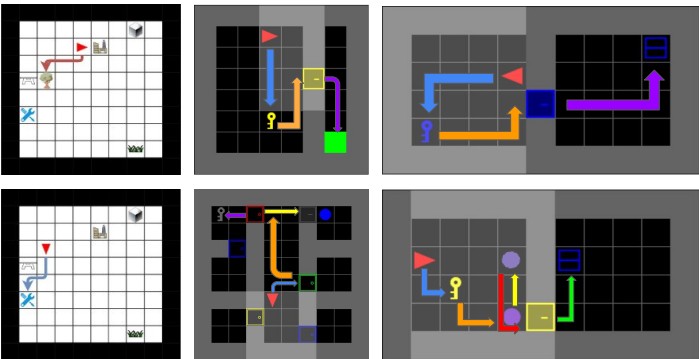

Figure 2: Environments in Craft (Lu et al., 2021) and Minigrid (Chevalier-Boisvert et al., 2018) used for sequence decomposition. Ground-truth sub-routine action sequences are represented with coloured arrows.

`DoorKey-8x8`, `UnlockPickup-v0`, `BlockedUnlockPickup` and `KeyCorridor-S4R3`. Episodes in this Minigrid are procedurally generated, leading to far greater variability for the actions and observations that constitute a sub-routine compared to in Craft. Further, in Minigrid environments, two actions (`PICKUP` or `TOGGLE`) typically indicate the presence of sub-routine boundaries as opposed to in Craft where this is marked by only a single `USE` action. In Figure 2, we can see some representative trajectories from these environments with the coloured arrows indicating the various ground-truth sub-routines.

## 4.2 Evaluation Metrics

To quantitatively measure the quality of the action sequence decomposition, we use the F1 score (with a tolerance of 1 consistent with Lu et al. (2021)) based on the accuracy of the boundary index predictions with respect to ground-truth. Further, we also report the alignment accuracy between the predicted and ground-truth sub-routines consistent with prior work (Lu et al., 2021). Please refer to Appendix A.1 for further details on how ground truth sub-routine boundaries are computed.

## 4.3 Hyperparameter Search

We report results after conducting an extensive hyperparameter search (up to 300 configurations) for each method. We include training parameters like batch size and learning rate, capacity parameters like layer sizes, and model-specific parameters such as the number of Slot Attention iterations in case of SloTTAr.

General trends we encountered as part of this search for SloTTAr are that (1) using only a single Transformer layer (for the Transformer Encoder and Decoder modules) was necessary to limit the tendency of the self-attention layer to aggregate information across non-contiguous temporal blocks; (2) that the capacity of the slots must be sufficiently bottlenecked since otherwise the model tends to only use a single slot to model the entire sequence; and (3) that often a single iteration of Slot Attention was sufficient to learn the decomposition for all the environments. In fact, performance usually degraded slightly with additional iterations, as is consistent with prior work (Locatello et al., 2020; Kipf et al., 2022).

Below we report results only for the best configurations of SloTTAr and the baseline models (CompILE, OMPN) with mean and standard deviation computed over 5 seeds. Details about the search parameters for each method and the best configurations can be found in Appendix A.3.

Table 1: F1 scores and alignment accuracies on Craft (Lu et al., 2021) (fully and partially observable).

|  | Craft (fully) | | Craft (partial) | |
|---|---|---|---|---|
|  | F1 score | Align. acc. | F1 score | Align. acc. |
| CompILE | 83.10 (3.07) | 85.77 (0.51) | 70.74 (14.14) | 67.71 (16.81) |
| OMPN | 98.49 (0.16) | 97.32 (0.33) | **95.84 (0.71)** | **93.68 (0.96)** |
| OMPN-2 | 97.04 (1.03) | 94.58 (2.44) | 91.20 (2.13) | 85.04 (3.82) |
| SloTTAr | **99.84 (0.02)** | **99.41 (0.25)** | 80.51 (4.47) | 83.14 (3.87) |

Table 2: F1 scores and alignment accuracies on partially observable versions of Minigrid environments – `DoorKey-8x8` and `UnlockPickup-v0`.

|  | DoorKey-8x8 (partial) | | UnlockPickup-v0 (partial) | |
|---|---|---|---|---|
|  | F1 score | Align. acc. | F1 score | Align. acc. |
| CompILE | 43.78 (2.92) | 64.93 (2.96) | 43.81 (1.70) | 69.85 (2.01) |
| OMPN | 45.03 (14.04) | 57.20 (4.35) | 55.27 (5.76) | 62.08 (7.78) |
| OMPN-2 | 48.08 (10.48) | 60.29 (8.47) | 48.47 (14.39) | 46.25 (6.84) |
| SloTTAr ($\beta$=0) | 72.19 (9.87) | 63.93 (16.14) | 75.08 (8.78) | 76.64 (7.14) |
| SloTTAr | **91.73 (3.75)** | **93.96 (0.96)** | **79.08 (8.01)** | **81.63 (7.41)** |

### 4.4 Results

#### 4.4.1 Craft

Table 1 shows the results of comparing SloTTAr to the baseline models on environments in Craft. All trajectories in these environments have 4 sub-routines executed. On the fully observable Craft tasks, it can be seen how SloTTAr outperforms both CompILE and OMPN. Similarly, on the partially observable settings in Craft, it can be seen how SloTTAr outperforms CompILE, although this time it performs worse compared to OMPN. We note that the best configuration for OMPN in Table 1 was obtained when using 3 levels of hierarchy depth (Table 17), and speculate how the strong hierarchical inductive bias in OMPN gives it an edge in this setting as it closely reflects the hierarchical sub-routine structure in Craft. Indeed, when we only consider OMPN configurations having two levels of memory (OMPN-2) it can be observed how the difference in alignment accuracy to the best performing configuration greatly reduces.

#### 4.4.2 Minigrid – DoorKey-8x8 and UnlockPickup

All trajectories from these environments involve executing 3 sub-routines. Table 2 shows that on these harder `DoorKey-8x8` and `UnlockPickup-v0` partially observable Minigrid environments, SloTTAr significantly outperforms both CompILE and OMPN in terms of both F1 and alignment accuracy[2]. We also show results from an ablation where the KL term is removed from the loss by setting $\beta = 0$ (in Equation (3)) to emphasize the role of the empirical prior in ascertaining the correct number of "active" slots to be used.

We investigated whether the reduced performance for OMPN on the Minigrid environments is due to these datasets using multiple delimiting tokens. Table 19 in Appendix A.6 reports the performance for synthetic versions of the `DoorKey-8x8` and `UnlockPickup` datasets with a single delimiting token (`TOGGLE`) instead of the default two i.e., `TOGGLE` and `PICKUP`. We see that the quality of decomposition by OMPN improves modestly but a significant gap to SloTTAr remains.

---

[2]In a preliminary version of this work we reported an F1 score of 50.58 (4.01) and an alignment accuracy of 72.88 (2.58) on `DoorKey-8x8` (partial) for CompILE due to an inconsistency in how the action sequence was pre-processed (refer to Appendix A.6 for further details). This did not affect our findings that SloTTAr significantly outperforms CompILE on this environment.

### 4.4.3   Minigrid – BlockedUnlockPickup and KeyCorridor

All trajectories in `BlockedUnlockPickup` involve executing either 4 or 5 sub-routines and in `KeyCorridor-S4R3` between 4 to 9 sub-routines. Table 3 shows the performance of SloTTAr against baselines on `BlockedUnlockPickup` and `KeyCorridor-S4R3` environments which have variable number of sub-routines. The baseline models (i.e., CompILE and OMPN) receive supervision on the number of sub-routines in every trajectory during training and/or evaluation. In contrast, SloTTAr performs the decomposition in a mostly unsupervised manner and only uses the empirical *distribution* of the number of sub-routines across the training dataset as a prior during training.

We can see that despite this advantage given to the baseline models, SloTTAr still significantly outperforms CompILE on both environments and remains competitive with OMPN. The decomposition quality of all these models on the `KeyCorridor-S4R3` environment is rather poor. This result partially serves to highlight the limits of current state-of-the-art approaches to decompose action trajectories and recover semantically meaningful sub-routine parts in a mostly unsupervised manner.

Table 3: F1 scores and alignment accuracies on Minigrid environments with variable number of sub-routines per-trajectory – `BlockedUnlockPickup` and `KeyCorridor-S4R3`. CompILE and OMPN receive supervision on the number of sub-routines in every trajectory during training and/or evaluation (denoted by "oracle").

|  | BlockedUnlockPickup (partial) | | KeyCorridor-S4R3 (partial) | |
| --- | --- | --- | --- | --- |
|  | F1 score | Align. acc. | F1 score | Align. acc. |
| CompILE (oracle) | 19.99 (1.73) | 18.15 (1.11) | 0.00 (0.00) | 40.37 (0.01) |
| OMPN (oracle) | **55.80 (11.31)** | 45.83 (8.73) | **48.86 (18.34)** | **41.58 (2.64)** |
| SloTTAr | 41.44 (3.54) | **63.48 (0.13)** | 48.81 (5.81) | 37.98 (6.04) |

### 4.4.4   Analysis

To better understand the performance of SloTTAr and verify that global access to input sequence is beneficial, we quantitatively measure the extent to which slots access information from past and future sequence tokens when modeling a particular sub-routine. We introduce two metrics we call "Backward Access" (BA) and "Forward Access" (FA), which are computed for a single trajectory as follows:

$$
\text{BA} = \frac{1}{K}\sum_{k=1}^{K} \frac{\sum_{l=1}^{\alpha_{min}-1} \mathbb{1}(\texttt{slot\_attn\_k}_l > \texttt{t}_{on})}{\alpha_{min}-1} \quad \text{where} \quad \alpha_{min} = \min\big\{1 \leq l \leq L : \texttt{mask\_k}_l > \texttt{t}_{on}\big\}
$$

$$
\text{FA} = \frac{1}{K}\sum_{k=1}^{K} \frac{\sum_{l=\alpha_{max}+1}^{L} \mathbb{1}(\texttt{slot\_attn\_k}_l > \texttt{t}_{on})}{L - \alpha_{max}+1} \quad \text{where} \quad \alpha_{max} = \max\big\{1 \leq l \leq L : \texttt{mask\_k}_l > \texttt{t}_{on}\big\}
$$

(4)

where $\mathbb{1}()$ is an indicator function and $\texttt{t}_{on} \in (0,1)$ denotes a threshold for activation (here $\texttt{t}_{on} = 0.8$). For the degenerate case where $\alpha_{min} = 1$ or it is an empty set, the BA is set to zero. Likewise, when $\alpha_{max} = L$ or is an empty set the FA is set to zero. Intuitively, these metrics measure the fraction of total timesteps accessed by each slot `slot_k` that lie in the past or future as indicated by the corresponding slot attention weights over the input sequence (`slot_attn_k`) exceeding the threshold $\texttt{t}_{on}$. To determine the boundary start ($\alpha_{min}$) and end ($\alpha_{max}$) indices of the segment modeled by `slot_k`, we binarize its associated decoder mask `mask_k` and threshold it in the same way.

Table 4 shows these metrics for SloTTAr on the test split of 4 environments namely — fully and partially observable variants of Craft, `DoorKey-8x8` and `UnlockPickup-v0`. It can be seen how the slots in our fully parallel model learn to effectively utilize information from past and future sequence tokens to solve the sequence decomposition task.

Table 4: Forward Access Backward Access metrics (mean and standard deviation over 5 seeds) for SloTTAr on 4 datasets – Craft (fully and partially observable), DoorKey-8x8 and UnlockPickup-v0.

|  | Forward Access | Backward Access |
| --- | --- | --- |
| Craft (fully) | 16.94 (6.63) | 9.93 (6.76) |
| Craft (partial) | 21.17 (1.75) | 9.44 (2.97) |
| DoorKey-8x8 | 26.85 (8.69) | 8.10 (8.77) |
| UnlockPickup-v0 | 15.94 (3.76) | 6.06 (3.71) |

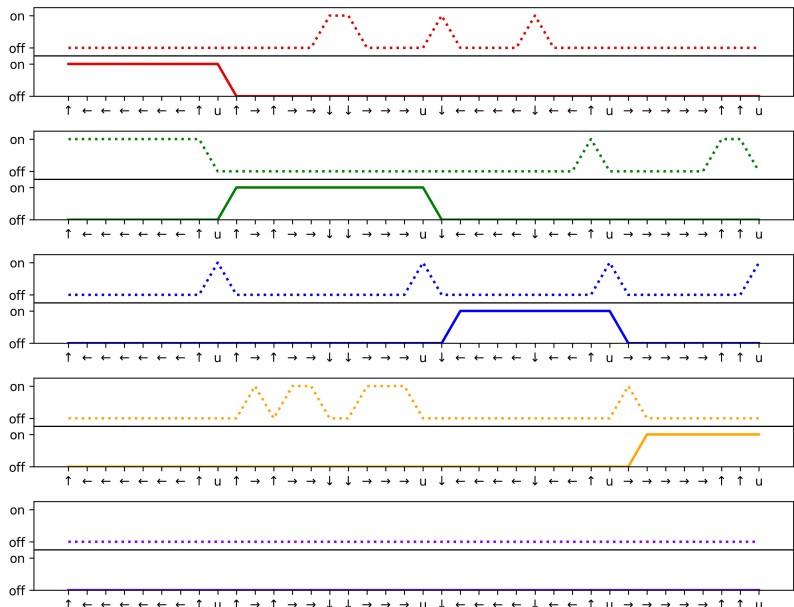

Figure 3: Within each subplot, the bottom panel shows thresholded alpha mask (in solid lines) and the top panel shows the thresholded slot attention weights (in dotted lines) on the y-axis for a sample trajectory from fully observable Craft environment against actions on the x-axis. Different subplots correspond to different slots (colour coded). We can see how each slot learns (via attention over the entire sequence) to gather contextual information (from both the left and right neighbourhood) in order to model its respective sub-sequence.

Qualitatively, in Figure 3 we show a representative visualization of the slot attention weights and alpha masks (`mask_k`) on a sample trajectory from the Craft fully observable environment (delimiters are the `USE` action 'u'). It can be seen that the 'active' slots (first four rows) have learned to gather information to the left (past) and to the right (future) of the sub-routine segment it models to acquire the necessary contextual information needed to perform this decomposition. Please refer to Appendix A.5 for additional visualizations of sample trajectories from other environments.

### 4.4.5 Computational Efficiency

A further advantage of using a fully parallel architecture as in SloTTAr is the potential gain in speed. While, the use of Transformers yields a computational complexity that is quadratic in sequence length, all these computations can be performed in parallel. In Table 5 we report the number of tokens processed per-second for each model during training (forward and backward pass) and at testing time (forward pass). Additionally, we report the total wall clock time for training each model. It can be seen how SloTTAr is about 3x faster to train compared to CompILE and upto 7x faster to train compared to the OMPN model.

Table 5: First and second columns show the number of tokens (in thousand) processed per-second during training and testing on a single Nvidia GTX 1080Ti GPU card respectively. While, the third column shows the wall clock time for the model training to converge. These numbers are computed on the fully observable Craft task using a batch size of 64 and sequence length of 65 tokens. SloTTAr uses both hidden size and slot size of 128 units and 8 attention heads. The number of segments is 4. OMPN and OMPN-2 use 3 and 2 levels of memory hierarchy respectively. The hidden size of OMPN models is 128 and batch size of 128.

|         | Train | Test | Wall clock |
|---------|-------|------|------------|
| CompILE | 14    | 36   | 0h 47m     |
| OMPN    | 3     | 7    | 1h 46m     |
| OMPN-2  | 4     | 10   | 1h 18m     |
| SloTTAr | **46**| **114**| **0h 14m** |

### 4.4.6 Limitations

The focus of this work is on the fundamental problem of learning "meaningful" sub-routines with an emphasis on modularity and compositionality as the desirable properties used to learn these temporal abstractions. The applications of discovered sub-routines to actual online decision making (e.g., in hierarchical reinforcement learning) are left for future work. In fact, our current model design does not allow an out-of-the-box application for an RL setting. This is because currently our Decoder module requires global context (past and future timesteps) to output action logits. However, we can consider potential remedies and strategies to address this limitation. The most straight-forward approach is to train a separate auto-regressive model (e.g. an auto-regressive Transformer) on the discovered sub-routine state/action sequences by using the corresponding slot representation as a sequence-start symbol embedding (and by appending an end-of-sequence token at the end of the sub-routine).

## 5 Conclusion

We have proposed SloTTAr, a novel approach to learning action segments belonging to modular sub-routines (suitable as 'options' (Sutton et al., 1999)) along with the number of such sub-routines in a mostly unsupervised fashion. Our approach draws insight from prior literature on learning about visual objects (Locatello et al., 2020), Transformers (Vaswani et al., 2017) and adaptive computation (Banino et al., 2021) to improve over existing purely sequential approaches. We found that SloTTAr outperforms CompILE and OMPN in terms of recovering ground-truth sub-routine segments across a wide range environments in Craft (Lu et al., 2021) and Minigrid (Chevalier-Boisvert et al., 2018) that have a fixed number of sub-routines. On more sophisticated environments with trajectories containing varying numbers of sub-routines, it could also be observed how SloTTAr outperforms CompILE and remains competitive with OMNP, despite these baseline models requiring ground-truth information about the number of sub-routines at the level of individual trajectories. At the same time, these results also indicated how current state-of-the-art approaches are still limited in their ability to decompose action sequences into semantically meaningful modular sub-routines. Our analysis revealed how SloTTAr leverages parallel access to the full sequence to perform sequence decomposition, which further leads to a substantial speed-up through leveraging parallel computation. In this way, we demonstrated how general principles of similarity-based grouping used to segment visual inputs are also relevant for grouping other input modalities, suggesting many additional avenues for future research.

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

# A    Experimental Details

## A.1    Datasets

**Craft**    The craft environment was initially introduced and re-implemented as a gym environment (Andreas et al., 2017; Lu et al., 2021). Demonstration data is collected from the rollouts of the heuristic bots (Lu et al., 2021). We use rollouts from each of the 3 tasks `MakeAxe`, `MakeBed` and `MakeShears` for both the fully observable and partial observable versions of these environments Lu et al. (2021) to report the results in Table 1. We use 10 000 trajectories for training and another separate 1000 trajectories each for creating validation and test splits. These are used for hyperparameter tuning and reporting evaluation metrics respectively.

**Minigrid**    We use the `DoorKey-8x8`, `UnlockPickup-v0`, `BlockedUnlockPickup` and `KeyCorridor-S4R3` environments from Minigrid (Chevalier-Boisvert et al., 2018) as additional datasets. These environments use procedural generation for each episode ensuring more variability in the action sequences that make up a sub-routine in comparison to Craft. We collect demonstration data by training an A2C agent (Mnih et al., 2016) on this environment until the policy receives an episodic return of $\geq 0.9$, which is close to optimal. We use 10 000 rollouts collected from this "expert" A2C agent (Mnih et al., 2016) as the training set and 1000 rollouts each for the validation and test splits used to tune hyperparameters and report evaluation metrics respectively.

**Ground-truth segment generation**    For the Craft environments, the `USE` action serves as a delimiter that marks the end of sub-routines. In the Minigrid environments, the `PICKUP` and `TOGGLE` actions serve as delimiters. For the synthetic versions of the `DoorKey-8x8` and `UnlockPickup` datasets with a single delimiter (used in section 4.4.2), we simply replace all occurrences of the delimiting action `PICKUP` with `TOGGLE`. We extract the ground-truth boundaries of sub-routines using the indices of these action tokens in the sequences as markers. Ground-truth sub-routine indices for every sequence are in ascending order from left-to-right where delimiting points are specified using the heuristics described above. Table 6 shows the ground-truth sub-routine types for each of the environments.

Table 6: Examples of ground-truth sub-routines for Craft (Lu et al., 2021) (fully & partially observable) , `DoorKey-8x8`, `UnlockPickup`, `BlockedUnlockPickup` and `KeyCorridor` environments from Minigrid suite (Chevalier-Boisvert et al., 2018).

|  |  |
|---|---|
| MakeAxe | get wood, make at workbench, get iron, make at toolshed |
| MakeBed | get wood, make at toolshed, get grass, make at workbench |
| MakeShears | get wood, make at workbench, get iron, make at workbench |
| DoorKey-8x8 | pickup key, open door, go to green goal |
| UnlockPickup-v0 | pickup key, open door, pickup box |
| BlockedUnlockPickup | pickup key, remove boulder, open door, pickup box |
| KeyCorridor | pickup key, open door, pickup ball |

## A.2    Evaluation Metrics

Evaluation metrics used to quantitatively measure the segmentation performance of models are the alignment accuracy of sub-routine prediction and F1 score of boundary index prediction and is consistent with prior work (Lu et al., 2021; Kipf et al., 2019).

**Alignment Accuracy**    The expression to compute alignment accuracy is shown below:

$$\text{Align Acc.} = \frac{1}{L * N} \sum_{n}^{N} \sum_{l}^{L} \mathbb{1}(\hat{s}_l^n = s_l^n)$$

where $N$ is the number of sequences, $L$ is the sequence length (possibly different for each sequence), $\hat{s}_l$ is the predicted sub-routine and $s_l$ is the ground-truth sub-routine action $a_l$ belongs to in the $n^{\text{th}}$ sequence.

**F1 Score**   The F1 score on predicted boundary indices is computed as shown below:

$$\text{F1 score} = \frac{2 \times \text{precision} \times \text{recall}}{\text{precision} + \text{recall}}$$

where precision is computed as,

$$\text{precision} = \frac{\# \text{ matches of boundary predictions with ground-truth}}{\text{total} \# \text{ of boundary predictions}}$$

and recall is computed as,

$$\text{recall} = \frac{\# \text{ matches of boundary predictions with ground-truth}}{\text{total} \# \text{ ground-truth boundaries}}$$

Please refer to the Appendix D.1 in prior work (Kipf et al., 2019) for further details.

### A.3   Architecture and Training Details

**Input Processing**   The `Embedding` layer used (Section 2) maps action tokens $a_l$ to $D_{enc}$ dimensions (denoted by hidden size in Table 9 and Table 10). The `Linear` layer used (Section 2) maps observations $\boldsymbol{o}_l$ to $D_{enc}$ dimensions. These $D_{enc}$ dimensional action and observation features are then concatenated and processed by a 1-layer MLP with $D_{enc}$ units and `ReLU` activation to learn a joint action-observation embedding space.

**Encoder**   The linear layers used in self-attention have $D_{enc}$ units in the `Encoder` block. The position-wise feedforward network used in the `Encoder` uses 4 times the number of units as hidden size. We add the standard sinusoidal positional encoding $\boldsymbol{p}_{\text{sin}}$ (Vaswani et al., 2017) to the joint action-observation features $\boldsymbol{z_{ao}}$. Further, we use a linear layer to generate the learned positional encoding $\boldsymbol{p}_{\text{learn}}$ that is added to the outputs $\boldsymbol{h}$ from the `Encoder` module. The final configuration(s) for each of these hyperparameters are shown in Table 9 and Table 10 for fixed and variable slot environments respectively. The hyperparameter sweep configurations are shown in Table 7 and Table 8 for fixed and variable slot environments respectively. The overall `Encoder` block uses the same architectural template as the Transformer encoder (Vaswani et al., 2017).

**Slot Attention**   The slot attention module uses linear layers with $D_{slots}$ units to generate keys and values. The recurrent update function is implemented using a GRU (Cho et al., 2014) (see also earlier work (Gers et al., 2000)) with $D_{slots}$ units (denoted by slot size in Table 9 and Table 10). The MLP network used for the residual update is implemented using a 2-layer network with $D_{slots}$ units and ReLU and no activation for the hidden layer and output layers respectively. Further, the initial query vector for `slot_k` is sampled from a Gaussian distribution with mean 0 and standard-deviation $\sigma$.

**Decoder**   The decoder architecture adapts the spatial broadcast decoder (Watters et al., 2019) in a suitable manner to decode all slots into their corresponding action segments. We broadcast the slot representations along the sequence length $L$ and pass the observations $\boldsymbol{o_l}$ as inputs at each timestep to the `Decoder`. The Decoder module has the exact same architecture as the Transformer encoder with the exception that an additional output linear layer of $A + 1$ (where $A$ is the size of action space) dimensions is used to decode the slot representations into predicted action logits and the end position logits. Segment masks are computed given end position logits as shown in Algorithm 2. Then we composite segment masks and predicted action logits to generate the full predicted action sequence.

**Hyperparameter Tuning**   We perform a random search of 300 hyperparameter configurations from all possible configurations in the hyperparameter sweeps shown in Table 7 and Table 8 on 3 seeds for fixed and variable slot environments respectively. We train our model for a maximum of 100 epochs on all environments

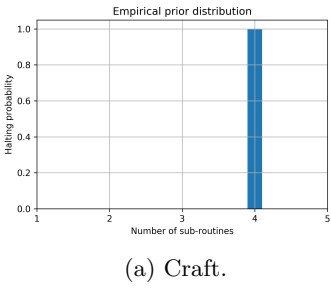

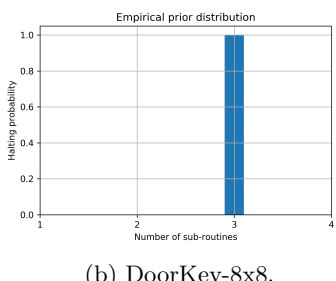

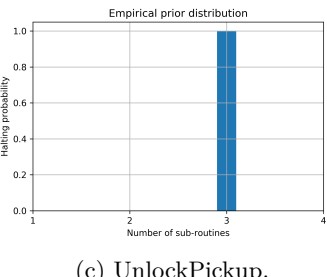

(a) Craft.  (b) DoorKey-8x8.  (c) UnlockPickup.

Figure 4: Empirical piece-wise priors for the Craft (fully and partially observable), Doorkey-8x8 and UnlockPickup environments in Minigrid all of which contains fixed number of sub-routines across trajectories.

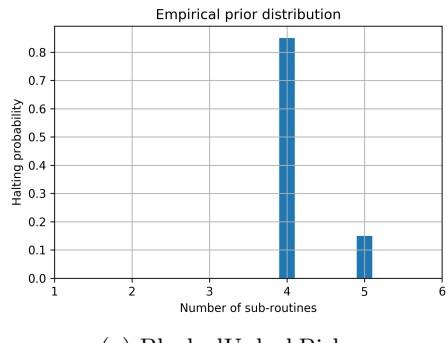

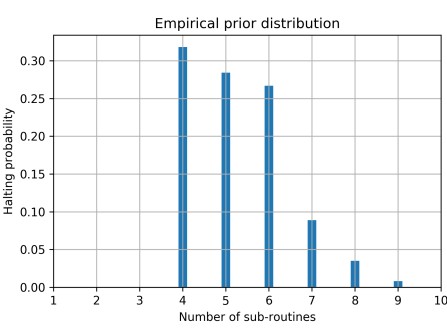

(a) BlockedUnlockPickup.  (b) KeyCorridor-S4r3.

Figure 5: Empirical discrete prior distributions for the BlockedUnlockPickup and KeyCorridor-S4R3 environments in Minigrid both of which contains variable number of sub-routines across trajectories.

with early stopping using the evaluation metrics (average over F1 and Align. acc.) on the validation set. Then, we picked the best performing configuration(s) for all of the environments for our model based on the evaluation metrics (average of F1 and Align. acc.) on the validation set and ran them again using 5 seeds to report final scores (shown in Table 1, Table 2, Table 3). The final configurations used for our model are shown in Table 9 and Table 10 for fixed and variable number of sub-routine environments respectively.

**Empirical Prior Distribution**   For the empirical prior distribution used in the PonderNet loss Equation (3), we use a categorical distribution over the maximum number of slots $K$. We compute the categorical probabilities as ratio of counts of the number of trajectories which have $k \in \{1, ..., K\}$ sub-routines and the total number of trajectories in the training dataset. Please refer to Figure 4 and Figure 5, which show these categorical priors for various datasets.

Since we have a discrete distribution for our prior and posteriors, we directly compute the KL-divergence between the halting probability ($p_{\text{halt}}$) and prior distribution ($p_e$) as shown below:

$$D_{KL}[p_{\text{halt}}||p_{\text{e}}] = \sum_{k'=1}^{K} p_{\text{halt}}(k') \log \left( \frac{p_{\text{halt}}(k')}{p_{\text{e}}(k') + \epsilon} \right) \tag{5}$$

where $\epsilon$ is a small positive number (here set to $1e-4$) for numerical stability. The maximum number of slots $K$ used for each of the environments can be seen in Figure 4 and Figure 5.

Table 7: Hyperparameter sweep configurations for SloTTAr on for Craft (Lu et al., 2021) (fully & partially observable), `DoorKey-8x8` and `UnlockPickup` environments from Minigrid (Chevalier-Boisvert et al., 2018). All of these environments have a fixed number of sub-routines across all trajectories.

| SloTTAr | Craft (fully) | Craft (partial) | DoorKey-8x8 | UnlockPickup |
|---|---|---|---|---|
| batch size | [32, 64, 128] | [32, 64, 128] | [32, 64, 128] | [32, 64, 128] |
| beta | [0.1, 0.5, 1.0] | [0.1, 0.5, 1.0] | [0.1, 0.5, 1.0] | [0.1, 0.5, 1.0] |
| hidden size | [64, 96, 128] | [64, 96, 128] | [64, 96, 128] | [64, 96, 128] |
| number of heads | [4, 8, 16] | [4, 8, 16] | [4, 8, 16] | [4, 8, 16] |
| slot size | [64, 96, 128] | [64, 96, 128] | [64, 96, 128] | [64, 96, 128] |
| slot std_dev | [0.5, 1.0] | [0.5, 1.0] | [0.5, 1.0] | [0.5, 1.0] |
| number of iterations | [1, 2] | [1, 2] | [1, 2] | [1, 2] |
| learning rate | [2.5e-4, 5e-4 1e-3] | [2.5e-4, 5e-4 1e-3] | [2.5e-4, 5e-4 1e-3] | [2.5e-4, 5e-4 1e-3] |

Table 8: Hyperparameter sweep configurations for SloTTAr for `BlockedUnlockPickup` and `KeyCorridor-S4R3` Minigrid environments (Chevalier-Boisvert et al., 2018). These environments have a variable number of sub-routines across different trajectories.

| SloTTAr | BlockedUnlockPickup | KeyCorridor-S4R3 |
|---|---|---|
| batch size | [32, 64, 128] | [32, 64, 128] |
| beta | [0.05, 0.1, 0.2] | [0.1, 0.2, 0.5] |
| hidden size | [64, 96, 128] | [64, 96, 128] |
| number of heads | [4, 8, 16] | [4, 8, 16] |
| slot size | [64, 96, 128] | [64, 96, 128] |
| slot std_dev | [0.5, 1.0] | [0.5, 1.0] |
| number of iterations | [1, 2] | [1, 2] |
| learning rate | [2.5e-4, 5e-4 1e-3] | [2.5e-4, 5e-4 1e-3] |

Table 9: Final hyperparameter configurations for SloTTAr for Craft (Lu et al., 2021) (fully & partially observable), `Doorkey-8x8`, `UnlockPickup` environments from Minigrid (Chevalier-Boisvert et al., 2018).

| SloTTAr | Craft (fully) | Craft (partial) | (DoorKey-8x8) | UnlockPickup |
|---|---|---|---|---|
| batch size | 64 | 64 | 32 | 32 |
| beta | 0.1 | 0.5 | 0.1 | 1.0 |
| hidden size | 128 | 128 | 128 | 128 |
| number of heads | 8 | 16 | 8 | 8 |
| number of layers | 1 | 1 | 1 | 1 |
| slot size | 128 | 64 | 128 | 128 |
| slot std_dev | 1.0 | 1.0 | 1.0 | 1.0 |
| number of iterations | 1 | 1 | 1 | 1 |
| number of slots | 5 | 5 | 4 | 4 |
| number of segments | 4 | 4 | 3 | 3 |
| optimizer | Adam | Adam | Adam | Adam |
| learning rate | 0.0005 | 0.0005 | 0.0005 | 0.0005 |

Table 10: Final hyperparameter configurations for SloTTAr for `BlockedUnlockPickup` and `KeyCorridor-S4R3` Minigrid environments (Chevalier-Boisvert et al., 2018). These environments have a variable number of sub-routines across different trajectories.

| SloTTAr | BlockedUnlockPickup | KeyCorridor-S4R3 |
|---|---|---|
| batch size | 64 | 32 |
| beta | 0.05 | 0.5 |
| hidden size | 96 | 96 |
| number of heads | 8 | 8 |
| number of layers | 1 | 1 |
| slot size | 96 | 96 |
| slot std_dev | 1.0 | 1.0 |
| number of iterations | 1 | 2 |
| number of slots | 5 | 10 |
| number of segments | 4-5 | 4-9 |
| optimizer | Adam | Adam |
| learning rate | 0.0005 | 0.00025 |

### A.4  Baseline Models and Training Details

We re-implemented CompILE [3] and OMPN [4] by adapting the original authors' implementations available on GitHub. We perform a random search of up to 300 hyperparameter configurations from all possible configurations in the hyperparameter sweeps shown in Table 11, Table 12, Table 15 and Table 16 for all environments and both baseline models (CompILE and OMPN) with 3 seeds. Then, we picked the best performing configuration(s) based on their evaluation metrics on the validation set from this sweep. We run these best performing configuration(s) on 5 seeds to obtain all results shown in Table 1, Table 2 and Table 3. We checkpoint models during the course of training based on their evaluation metric scores on the validation set. We use the best checkpoints to report the final scores shown in Table 1, Table 2 and Table 3 for all the baseline models.

The final configurations used for the baseline models (CompILE and OMPN) on the 3 environments are shown in Table 13, Table 14, Table 17 and Table 18 for fixed and variable slot environments.

Table 11: Hyperparameter sweep configurations for CompILE for Craft (Lu et al., 2021) (fully & partially observable), `Doorkey-8x8` and `UnlockPickup` environments from Minigrid (Chevalier-Boisvert et al., 2018). For the full description of all these hyperparameters, we refer the readers to Kipf et al. (2019).

| CompILE | Craft (fully) | Craft (partial) | DoorKey-8x8 | UnlockPickup |
|---|---|---|---|---|
| batch size | [64, 128, 256] | [64, 128, 256] | [32, 64, 128] | [32, 64, 128] |
| beta_b = beta_z | [0.01, 0.05, 0.1] | [0.01, 0.05, 0.1] | [0.01, 0.05, 0.1] | [0.01, 0.05, 0.1] |
| hidden size | [64, 128] | [64, 128] | [64, 128] | [64, 128] |
| latent dist. | ['gaussian', 'concrete'] | ['gaussian', 'concrete'] | ['gaussian', 'concrete'] | ['gaussian', 'concrete'] |
| latent size | [64, 128] | [64, 128] | [64, 128] | [64, 128] |
| learning rate | [5e-5, 2.5e-4 1e-3] | [5e-5, 2.5e-4 1e-3] | [1e-4, 2e-4 5e-4] | [1e-4, 2e-4 5e-4] |
| prior rate | [2, 3, 4] | [2, 3, 4] | [2, 3, 4] | [2, 3, 4] |

Table 12: Hyperparameter sweep configurations for CompILE for `BlockedUnlockPickup` and `KeyCorridor-S4R3` environments from Minigrid (Chevalier-Boisvert et al., 2018).

| CompILE | BlockedUnlockPickup | KeyCorridor-S4R3 |
|---|---|---|
| batch size | [32, 64, 128] | [32, 64, 128] |
| beta_b = beta_z | [0.05, 0.1, 0.2] | [0.01, 0.05, 0.1] |
| hidden size | [64, 128] | [64, 128] |
| latent dist. | ['gaussian', 'concrete'] | ['gaussian', 'concrete'] |
| latent size | [64, 128] | [64, 128] |
| learning rate | [5e-5, 2.5e-4 1e-3] | [5e-5, 2.5e-4 1e-3] |
| prior rate | [2, 3, 4] | [2, 3, 4] |

---

[3]https://github.com/tkipf/compile
[4]https://github.com/Ordered-Memory-RL/ompn_craft

Table 13: Final hyperparameter configurations for CompILE for Craft (Lu et al., 2021) (fully & partially observable), `DoorKey-8x8` and `UnlockPickup` environments from Minigrid Chevalier-Boisvert et al. (2018). All of these environments have a fixed number of sub-routines across all trajectories.

| CompILE | Craft (fully) | Craft (partial) | DoorKey-8x8 | UnlockPickup |
|---|---|---|---|---|
| batch size | 128 | 128 | 32 | 64 |
| beta_b | 0.01 | 0.01 | 0.01 | 0.05 |
| beta_z | 0.01 | 0.01 | 0.01 | 0.05 |
| prior rate | 3 | 3 | 4 | 2 |
| hidden size | 128 | 128 | 128 | 64 |
| latent dist. | 'gaussian' | 'gaussian' | 'gaussian' | 'gaussian' |
| latent size | 128 | 128 | 128 | 64 |
| number of segments | 4 | 4 | 3 | 3 |
| optimizer | Adam | Adam | Adam | Adam |
| learning rate | 0.00025 | 0.00025 | 0.0005 | 0.0002 |

Table 14: Final hyperparameter configurations for CompILE for `BlockedUnlockPickup` and `KeyCorridor-S4R3` environments from Minigrid Chevalier-Boisvert et al. (2018). All of these environments have a fixed number of sub-routines across all trajectories.

| CompILE | BlockedUnlockPickup | KeyCorridorS4R3 |
|---|---|---|
| batch size | 128 | 128 |
| beta_b | 0.1 | 0.1 |
| beta_z | 0.1 | 0.1 |
| prior rate | 3 | 3 |
| hidden size | 128 | 128 |
| latent dist. | 'gaussian' | 'gaussian' |
| latent size | 128 | 128 |
| number of segments | 4-5 | 4-9 |
| optimizer | Adam | Adam |
| learning rate | 0.00025 | 0.00025 |

Table 15: Hyperparameter sweep configurations for OMPN for Craft (Lu et al., 2021) (fully & partially observable) , `Doorkey-8x8` and `UnlockPickup` environments from Minigrid (Chevalier-Boisvert et al., 2018). All of these environments have a fixed number of sub-routines across all trajectories.

| OMPN | Craft (fully) | Craft (partial) | DoorKey-8x8 | UnlockPickup |
|---|---|---|---|---|
| batch size | [64, 128, 256] | [64, 128, 256] | [32, 64, 128] | [32, 64, 128] |
| hidden size | [64, 128] | [64, 128] | [64, 128] | [64, 128] |
| learning rate | [1e-4, 2.5e-4, 5e-4] | [1e-4, 2.5e-4, 5e-4] | [1e-4, 2e-4, 5e-4] | [1e-4, 2e-4, 5e-4] |
| number of slots | [2, 3] | [2, 3] | [2, 3] | [2, 3] |
| max. gradient norm | [0.5, 1.0, 2.0] | [0.2, 0.5, 1.0] | [0.2, 0.5, 1.0] | [0.5, 1.0, 2.0] |

Table 16: Hyperparameter sweep configurations for OMPN for `BlockedUnlockPickup` and `Doorkey-8x8` environments from Minigrid (Chevalier-Boisvert et al., 2018). All of these environments have a variable number of sub-routines across all trajectories.

| OMPN | BlockedUnlockPickup | KeyCorridor-S4R3 |
|---|---|---|
| batch size | [32, 64, 128] | [32, 64, 128] |
| hidden size | [64, 128] | [64, 128] |
| learning rate | [1e-4, 2e-4, 5e-4] | [1e-4, 2e-4, 5e-4] |
| number of slots | [2, 3, 4] | [2, 3, 4] |
| max. gradient norm | [0.2, 0.5, 1.0] | [0.2, 1.0, 2.0] |

Table 17: Final hyperparameter configurations for OMPN for Craft (Lu et al., 2021) (fully & partially observable), `Doorkey-8x8` and `UnlockPickup` environments from Minigrid (Chevalier-Boisvert et al., 2018). All of these environments have a fixed number of sub-routines across all trajectories.

| OMPN | Craft (fully) | Craft (partial) | DoorKey-8x8 | UnlockPickup |
|---|---|---|---|---|
| batch size | 128 | 128 | 64 | 64 |
| hidden size | 128 | 128 | 128 | 64 |
| number of slots | 3 | 3 | 3 | 3 |
| number of segments | 4 | 4 | 3 | 3 |
| max. gradient norm | 1.0 | 1.0 | 1.0 | 2.0 |
| optimizer | Adam | Adam | Adam | Adam |
| learning rate | 0.00025 | 0.00025 | 0.0001 | 0.0002 |

Table 18: Final hyperparameter configurations for OMPN `BlockedUnlockPickup` and `KeyCorridor-S4R3` environments from Minigrid (Chevalier-Boisvert et al., 2018). These environments have a variable number of sub-routines across all trajectories.

| OMPN | BlockedUnlockPickup | KeyCorridor-S4R3 |
|---|---|---|
| batch size | 32 | 32 |
| hidden size | 128 | 128 |
| number of slots | 3 | 3 |
| number of segments | 4-5 | 4-9 |
| max. gradient norm | 0.2 | 2.0 |
| optimizer | Adam | Adam |
| learning rate | 0.0001 | 0.0001 |

### A.5 Visualizations

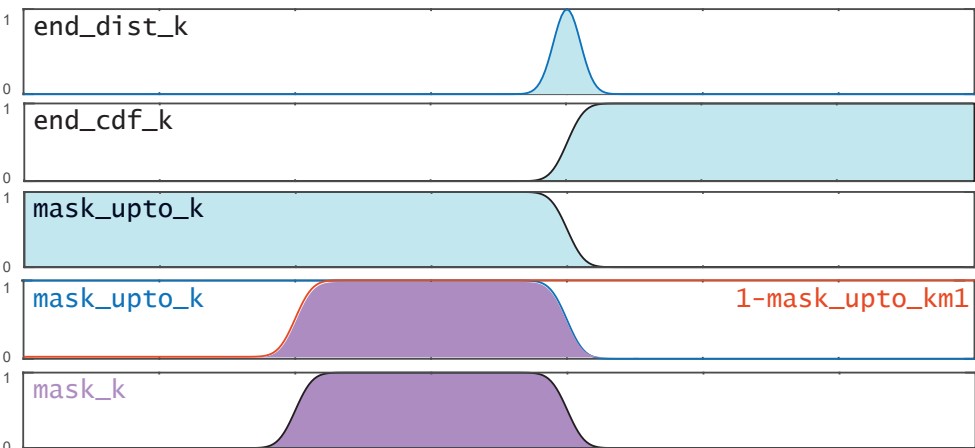

Figure 6: Panels 1-5 show a representative visualization of the outputs of computational steps 2-5 of Algorithm 2 respectively. In panel 4, curve in red shows the negation of the variable `mask_upto_km1` and curve in blue shows the variable `mask_upto_k`.

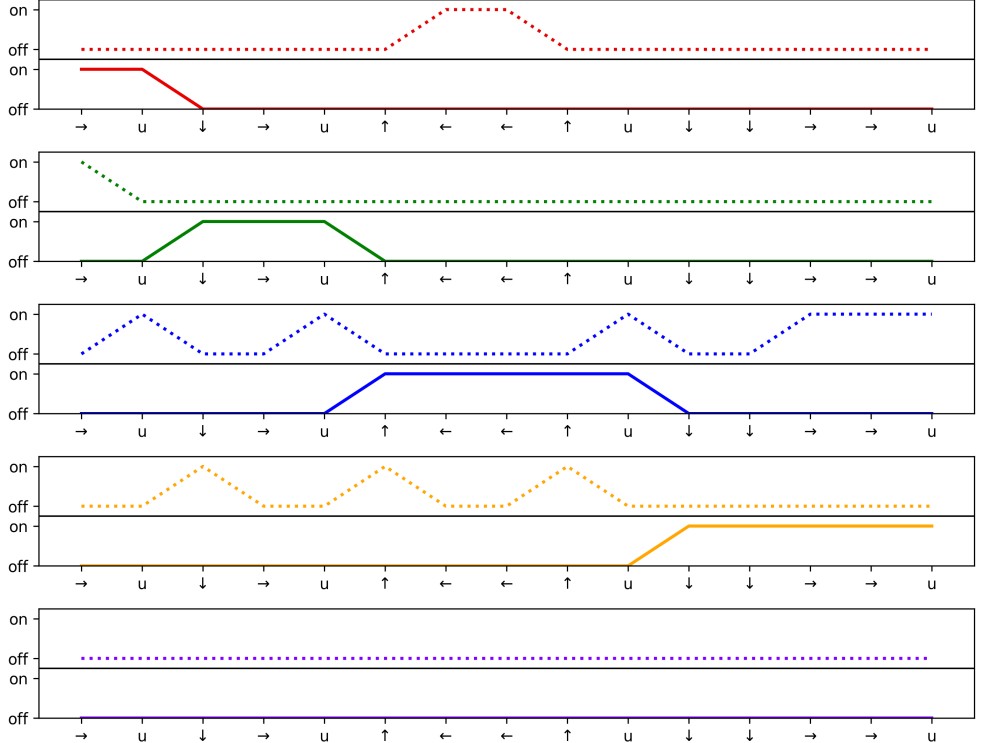

Figure 7: Within each subplot, the bottom panel shows thresholded alpha mask (in solid lines) and the top panel shows the thresholded slot attention weights (in dotted lines) on the y-axis for a sample trajectory from partially observable Craft environment against actions on the x-axis. Different subplots correspond to different slots (colour coded).

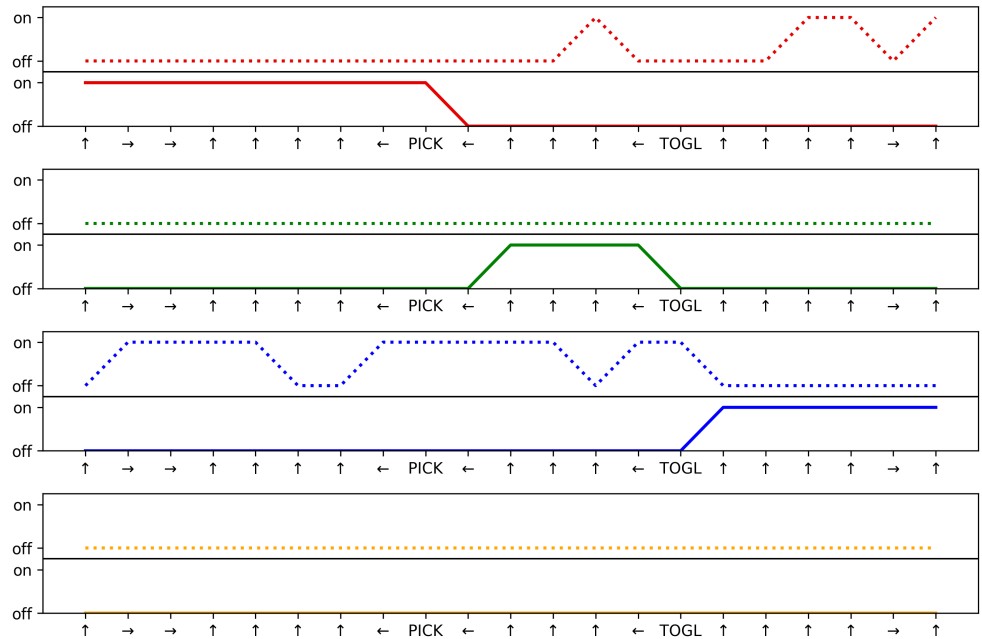

Figure 8: Within each subplot, the bottom panel shows thresholded alpha mask (in solid lines) and the top panel shows the thresholded slot attention weights (in dotted lines) on the y-axis for a sample trajectory from partially observable `DoorKey-8x8` Minigrid environment against actions on the x-axis. Different subplots correspond to different slots (colour coded).

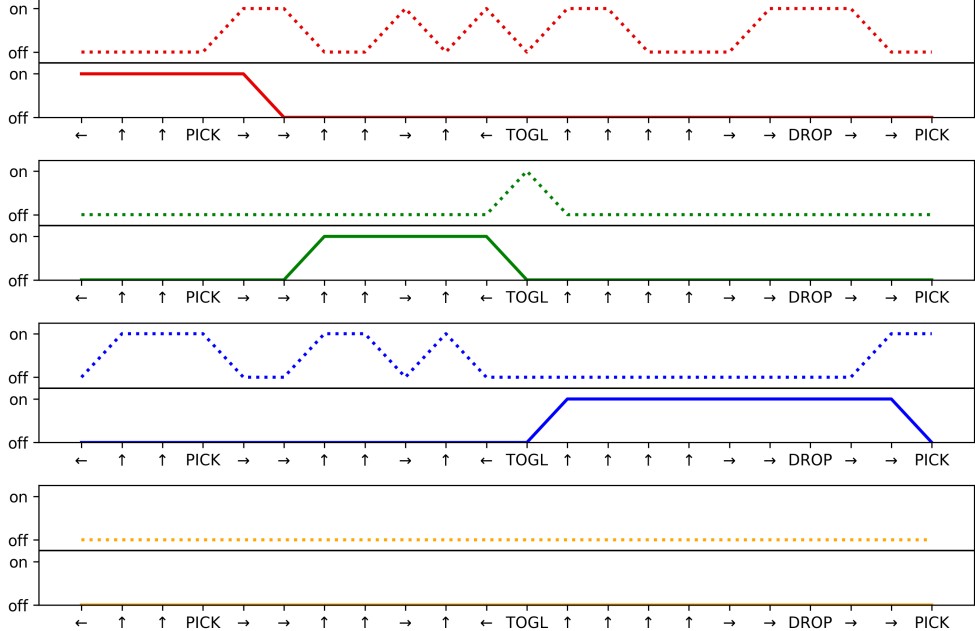

Figure 9: Within each subplot, the bottom panel shows thresholded alpha mask (in solid lines) and the top panel shows the thresholded slot attention weights (in dotted lines) on the y-axis for a sample trajectory from partially observable `UnlockPickup` Minigrid environment against actions on the x-axis. Different subplots correspond to different slots (colour coded).

### A.6 Additional Results

Table 19: F1 scores and alignment accuracies on the synthetic single delimiter versions of Minigrid environments – `DoorKey-8x8` and `UnlockPickup-v0`. F1 scores are computed with tolerance=1 consistent with Lu et al. (2021).

| single-delimiter | DoorKey-8x8 (partial) | | UnlockPickup-v0 (partial) | |
|---|---|---|---|---|
| | F1 score | Align. acc. | F1 score | Align. acc. |
| OMPN | 46.26 (13.99) | 56.11 (6.23) | 62.35 (18.39) | 70.83 (10.72) |
| OMPN-2 | 53.70 (12.01) | 65.55 (8.48) | 53.58 (19.57) | 59.84 (15.41) |

**Preprocessing Inconsistency.** We observed that the performance of CompILE improves on `DoorKey-8x8` (partial) Minigrid environment when the preprocessing of action sequences does not append a `DONE` token at the end. However, this is inconsistent with the preprocessing used by Kipf et al. (2019). Therefore for all CompILE results reported on Minigrid environments in this work, we maintain consistency with the preprocessing pipeline used by Kipf et al. (2019) and append a `DONE` token at the end of action sequences.

