# OpenReview forum: "Unsupervised Learning of Temporal Abstractions with Slot-based Transformers"
_TMLR — Rejected by TMLR_

### Review · Reviewer_oL8J · 2022-04-02

**Summary Of Contributions:**

This paper considers the problem of segmenting observation-action sequences into contiguous subsequences. The main idea is to make use of Transformers and Slot Attention to process the observation-action pairs in parallel, leading to more efficient computation, and both past and future attention without multiple sequential passes over the data. To circumvent the need for supervision on the number of segments per trajectory, the paper proposes to use the PonderNet loss function, which allows for an adaptive number of segments. Experiments in Craft and Minigrid consider the model’s segmentation ability compared to ground truth segmentations.

**Requested Changes:**

* If my understanding regarding the inapplicability of the model for decision making is correct, the paper should be reframed considerably to make this clear. In particular, the introduction and related work should emphasize that this work is really about event detection / behavior recognition. Through that lens, works on option discovery are much less relevant, and even “imitation learning” is a misnomer, because the model cannot actually imitate, it can only recognize.
* If my understanding on the above point is wrong, then the paper would be strengthened considerably with the inclusion of experiments where the learned models are used for decision making.
* I didn’t understand this: “Here, unlike in the original Slot Attention formulation, we additionally use separate shift and scaling parameters for each slot. This makes it possible to order them…” What is the relationship between those different parameters and the ordering? What prevents assigning a fixed order at the outset, and using the same shift and scaling parameters for all slots?
* Please make Figure 3 easier to interpret, either by adding description to the caption / main text, or by changing the visualization in some way (e.g., making it easier to distinguish the dashed line from the solid line when they sometimes overlap). Right now, I think the main takeaway is getting lost, at least for me.


### Minor

* In Algorithm 1, “:=” is used for updates, but not used anywhere else
* Algorithm 1 is nearly identical to Algorithm 1 from Locatello et al. (2020). It would be useful to color the parts that are different in this work (I think it should only be a few symbols)
* In Algorithm 2, mask_upto_k is not defined
* In Algorithm 2, there is a period in the comment on line 4, but not elsewhere
* My understanding is that the difference between OMPN and OMPN-2 in the results is that the former uses 3 levels of hierarchy depth, while the latter uses 2. For clarity, it may be better to use OMPN-3 and OMPN-2.

**Strengths And Weaknesses:**

* Strength: the analogy between segmentation in the image/object case and this temporal action case is nicely articulated.
* Strength: the paper is overall written very clearly. It is well organized, and I did not find serious grammatical issues or typos.
* Strength: the paper is clear about its relation to the closest prior works (CompILE and OMPN), and it makes a clear argument for why it should improve on those works. Moreover, the main experimental results support the argument that there is such an improvement for boundary detection.
* Strength: the figures, pseudocode, and notation are all very clear and helpful.
* Strength: the Hyperparameter Search section (4.3) is very nice in terms of transparency and thoroughness, and in combination with the extensive details in the appendix, could serve as a model for other papers.
* Strength: the computational efficiency results in Table 5 are clear and compelling.
* Strength/Weakness: I really like the idea behind the analysis in 4.4.4 with forward and backward access, and I appreciate the results in Table 4. However, I found the qualitative results in Figure 3 difficult to understand.
* Weakness: Much of the motivation for work in this direction, as discussed in the introduction, is to learn subroutines that facilitate decision making. However, it is not clear that the model proposed in this work could actually be used for decision making.
   * In the prior work on CompILE, the decoder takes the form of a latent-conditioned policy, which crucially depends on the observation. Here, it does not seem like the decoder depends on the observation, so no policies are learned, as far as I can tell.
   * It would not necessarily be accurate either to say that “macro-actions” are being learned. By analogy to slot attention, which “do not specialize to one particular type or class of object” (Locatello et al. 2020), it seems like the learned representations here would not specialize to one particular type or class of macro-action.
   * The parallel nature of the encoding has advantages for boundary detection, as the paper discusses, but it’s not clear how to use the model during online decision making, where at any given moment, only the past is known.
   * The absence of any experiments that use the learned models for decision making underscores this point.
* Weakness: there is precedent in the literature for using F1 scores and related metrics to evaluate models of this kind. However, I worry that there is a lot of subjectivity in the “ground truth” labels that underpin these results. Who is to say whether a pick-move-place is one subroutine, three subroutines, or some other number? (I would conjecture that there is much less agreement between humans about action trajectory segmentation versus object segmentation.) In my opinion, a better metric would evaluate the subroutines by using them for decision making, bringing me back to the point above.
* Weakness: one of the main points in the paper is regarding the use of the PonderNet loss. I do not think we have sufficient evidence that the use of this loss in this context is better than alternatives.
   * The authors argue that PonderNet reduces the necessary assumptions vs. prior work because only a distribution on the number of sequences is known, rather than the exact number.
   * But an obvious baseline/ablation would be to assume a large upper bound on the number of possible sequences and to use that instead. Assuming such a bound seems no more restrictive than assuming a distribution.
   * The argument that this work improves on prior work in this way is also somewhat undercut by the fact that a maximum number of slots must be assumed.

---

### Review · Reviewer_UQ4y · 2022-04-13

**Summary Of Contributions:**

This work presents a new method of identifying sub-routines (options) in a reinforcement learning context. The new method uses Transformers with Slot Attention. The design of the method permits greater parallelization compared to past approaches. Likewise, this method does not require as much supervision compared to baseline methods.

**Broader Impact Concerns:**

This work does not raise any ethical concerns.

**Requested Changes:**

No changes are critical to securing my recommendation.
More information about how the hyperpameter search was performed would strengthen the support of claims of this paper. For example, how were "possible configurations" chosen? The configurations considered in the appendix differ based on environment.

**Strengths And Weaknesses:**

Strengths:
- The claim that SloTTAr "is capable of outperforming strong baselines" is sufficiently supported. This demonstrates the relevance of this method and potential for future improvement.
- The claim that SloTTAr is ``up to 7x faster'' is also sufficiently supported. This demonstrates the parallelization benefits.
- The method is sufficiently explained and motivated well. The reduced need for supervision is clearly explained and demonstrated.

Weaknesses:
- The way that the hyperparameter search is performed for the results in the main body of the paper favors methods that are sensitive to hyperparameter choices (testing hundreds of configurations and reporting best performance).
- The proposed method still requires the empirical distribution of the number of sub-routines during training. This is not made sufficiently clear when initially describing SloTTAr's advantages over methods that require knowing the number of sub-routines within each trajectory. (More supervision is needed than is naturally available from expert demonstrations.)

---

### Review · Reviewer_yD5C · 2022-04-21

**Summary Of Contributions:**

Towards realizing intelligent agent’s ability in inferring the underlying abstraction structure of decision sequence, this work introduces a new architecture called SloTTAr, to learning sub-routines of decision sequence.

SloTTAr consists of Transformer with a Slot Attention Module for encoding and grouping, Spatial Broadcast Decoder for decoding, and PonderNet for implementing objective function. The whole architecture is trained through action sequence reconstruction in an end-to-end fashion.

Notably, SloTTAr requires a prior distribution of the number of sub-routines only in training, and is free of such prior knowledge during evaluation. This gains the superiority for SloTTAr in comparison with previous arts, and is called “in a mostly unsupervised fashion” in this work.

SloTTAr is evaluated and analyzed in partially and fully observable versions of 3 environments in Craft and on 4 partially observable environments in Minigrid.


**Requested Changes:**

- I recommend the authors to polish and improve the presentation. A few points are mentioned in $\textbf{Strengths and Weaknesses}$ part.
- I recommend the authors to take my advice and to amend the not ‘self-contained’ issue and add sufficient details of proposed methods.
- For experiments, I recommend the authors to refer to my detailed comments in $\textbf{Strengths and Weaknesses}$ part and provide more results and discussions.


**Strengths And Weaknesses:**

Strong Aspects:
+ This work studies the fundamental problem of learning to infer the underlying abstraction structure of decision sequence. The progress made in this direction may inspire many research fields such as Hierarchical Reinforcement Learning with Temporal Abstraction, unsupervised discovery of reusable and transferrable sub-skills (sub-routines).
+ The proposed, SloTTAr, manages to combining several existing methods and architectures into a unified architecture of convenient training and efficient computation, whose effeteness in predicting sub-routine is demonstrated.
+ SloTTAr works in a mostly unsupervised fashion, learning to the superiority upon the related work (i.e., CompILE and OMPN) considered in this work.

&nbsp;

Weak Aspects (Concerns & Questions):
- The presentation of the proposed approach is not clear enough and not self-contained. After referring to the original papers of the components encapsulated in SloTTAr and checking the appendix, I can understand most content of this paper.
  - As to ‘not self-contained’, ‘mask_upto_k’ and ‘CumSum’ in Algorithm 2 are not defined. This made me not able to correctly understand Algorithm 2. ‘slot_attn_k’ is not defined before Section 4.4.4, although I can understand this according to the context.
  - A modification made on Slot Attention Module used in this paper is “additionally use separate shift and scaling parameters ($\mu_k$ and $\sigma_k$) for each slot”. The paper says “This makes it possible to order them, as is needed during mask generation (Algorithm 2) and when computing the loss”. However, I did not find enough details and did not understand this.
  - Figure 1 does help yet it is not informative enough. I recommend the authors to annotate the figure with corresponding notations (e.g., slot_attn_k). Besides, Figure 1 is of low fidelity which can be improved by using vector diagram.
  - Overall, I think the presentation especially of Section 2 can be substantially improved.
- The novelty of the proposed architecture is limited since all components are existing methods and few modifications are made. For mask generalization (Algorithm 2) and active slot distribution (Eq.1 and 2) which is specific to SloTTAr, I have some questions below:
  - First, it may be too late to introduce $\lambda_k$ below Eq.1 as “We estimate this scalar probability $\lambda_k$ of halting … its respective slot representation slot$_k$”, and it is somewhat confusing because $\lambda_k$ is not defined before Eq.1. I think a proper place is the second paragraph of Section 2 where the outputs of the decoder are introduced.
  - I did not understand Algorithm 2 for the reason of non-self-contained presentation as mentioned above.
  - Although I did not understand the details of Algorithm 2, I can get the point of segment mask. I have a concern on the relationship between segment masks and active slots. How do they work together? Are the active slots determined by sampling from the distribution (defined in Eq.2) first followed by the mask generation; If so, how would it be when only 1 slot is active and the mask position of the slot is in the middle of sequence? Or, is the distribution defined in Eq.2 only used in establishing the objective function?
  -  For Slot Attention Module, it seems that there is no guarantee that each slot necessarily groups consecutive (state-)actions in the sequence which is ‘semantically valid’. Can the authors discuss more about this point?
  - Since the training of SloTTAr needs the prior distribution of the number of sub-routines, I think it is improper and misleading to use ‘Unsupervised Learning’ in the paper title.
  - For Eq.4, is it possible that the sets to obtain minimum/maximum are empty?
- For the experiements, I have following questions:
  - In Figure 3 and 6-8, it seems that the slot attention coefficients (in dashed lines) are binary. According to Algorithm 1 Line 4, it should be the output of Softmax. How to understand this?
  - According to Figure 4 and 5, the prior distribution is almost concentrated (i.e., deterministic) except on KeyCorridor-S4r3. Therefore, such prior knowledge used in training is close to providing the number of sub-routines. According to the performance of SloTTAr on KeyCorridor-S4r3 as shown in Table 3, I recommend the authors to conduct evaluation on more environments with varying number of sub-routines.
  - Another concern is that, one appealing advantage of SloTTAr is that it can decide the number of sub-routines adaptively. However, in fact, a maximum number ($K$) is needed. This means we have to set a proper $K$ in the target environments, will this raise a contradiction in the aforementioned advantage?
  - One more important perspective is that, I think the experiments should also contain the ablation study on including/not including the KL term in Eq.3. In addition, I think ablation studies on module choices, e.g., the possible alternatives of Slot Attention Module, Spatial Broadcast Decoder, are important.
  - Overall, I think the current experiments are not sufficient in the aspects of evaluation, ablation and discussion/analysis.

---

> ### Comment · Reviewer_UQ4y · 2022-04-25
> **TMLR Instructs to Disregard Novelty**
>
> Thank you for the thorough review!
>
> I had similar concerns about novelty and sources of performance improvements, but I noticed that TMLR specifically instructs not to consider novelty and instead focus on whether claims are supported. Likewise, the primary claim (that SloTTAr "is capable of outperforming strong baselines") is supported; I did not notice claims about which aspects of SloTTAr were responsible for performance.

---

> > ### Comment · Reviewer_yD5C · 2022-04-29
> > **Thank You for Reminding**
> >
> > Thank you for reminding! I've re-checked the TMLR guidelines for reviews.
> >
> > I will not take novelty in my acceptance/rejection recommendation later. As detailed in my comments, my major concerns are on the details of methodology and experiments. To me, these concerns are critical to assessing the technical soundness as well as the clarity of the narrative and arguments presented.

---

### Decision · Action_Editors · 2022-06-01

**Recommendation:** Reject

**Comment:**

This paper proposes a Transformer variation that is a fully parallel approach for processing trajectory data. The proposed method SloTTAr discovers subroutines without supervision. Reviewers appreciated these aspects of SloTTAr and also the fact that it proposes a general framework in which previous work can be cast as special instances. This can be useful for the field in general. However several concerns remained amongst the reviewers after a rich author-reviewer and reviewer-reviewer discussion period:

* How this work can be used in decision-making was not clear especially as this seems to be an important goal of the work. Reviewers suggest concrete usages in typical decision-making problems should at least be introduced and empirically demonstrated for didactic purposes.

* Presentation and organization of the paper can be improved substantially.

* SloTTAr still requires the empirical distribution of the number of sub-routines during training. This should be made more transparent early on as part of the presentation and organization.

* Empirical evaluation while it has improved from the initial submission, many reviewer points have not been adequately addressed as detailed in the reviewer-author discussions.

While the paper came quite close this time, it will be better for the authors to take more time and address the points above and resubmit. The review committee is of the opinion that this will lead to a stronger and more impactful paper.